# *Candida albicans* as an Essential “Keystone” Component within Polymicrobial Oral Biofilm Models?

**DOI:** 10.3390/microorganisms9010059

**Published:** 2020-12-28

**Authors:** Tracy Young, Om-Alkhir Alshanta, Ryan Kean, David Bradshaw, Jonathan Pratten, Craig Williams, Chris Woodall, Gordon Ramage, Jason L. Brown

**Affiliations:** 1School of Medicine, Dentistry and Nursing, College of Medical, Veterinary and Life Sciences, Glasgow University, Glasgow G12 8QQ, UK; t.young.1@research.gla.ac.uk (T.Y.); 2151874A@student.gla.ac.uk (O.-A.A.); 2Blutest Laboratories, 5 Robroyston Oval, Nova Business Park, Glasgow G33 1AP, UK; chris.woodall@blutest.com; 3Glasgow Biofilm Research Network, 378 Sauchiehall Street, Glasgow G2 3JZ, UK; Ryan.Kean@gcu.ac.uk (R.K.); craig.williams@mbht.nhs.uk (C.W.); 4Department of Biological and Biomedical Sciences, School of Health and Life Sciences, Glasgow Caledonian University, Glasgow G4 0BA, UK; 5Oral Healthcare R&D, GlaxoSmithKline Consumer Healthcare, Weybridge KT13 0DE, UK; david.j.bradshaw@gsk.com (D.B.); jonathan.r.pratten@gsk.com (J.P.); 6Microbiology Department, Lancaster Royal Infirmary, University of Lancaster, Lancaster LA1 4YW, UK

**Keywords:** fungal–bacterial interactions, oral biofilm, high-throughput, polymicrobial, antimicrobials, biofilm models

## Abstract

**Background**: Existing standardized biofilm assays focus on simple mono-species or bacterial-only models. Incorporating *Candida albicans* into complex biofilm models can offer a more appropriate and relevant polymicrobial biofilm for the development of oral health products. **Aims**: This study aimed to assess the importance of interkingdom interactions in polymicrobial oral biofilm systems with or without *C. albicans*, and test how these models respond to oral therapeutic challenges in vitro. **Materials and Methods**: Polymicrobial biofilms (two models containing 5 and 10 bacterial species, respectively) were created in parallel in the presence and absence of *C. albicans* and challenged using clinically relevant antimicrobials. The metabolic profiles and biomasses of these complex biofilms were estimated using resazurin dye and crystal violet stain, respectively. Quantitative PCR was utilized to assess compositional changes in microbial load. Additional assays, for measurements of pH and lactate, were included to monitor fluctuations in virulence “biomarkers.” **Results**: An increased level of metabolic activity and biomass in the presence of *C. albicans* was observed. Bacterial load was increased by more than a factor of 10 in the presence of *C. albicans*. Assays showed inclusion of *C. albicans* impacted the biofilm virulence profiles. *C. albicans* did not affect the biofilms’ responses to the short-term incubations with different treatments. **Conclusions**: The interkingdom biofilms described herein are structurally robust and exhibit all the hallmarks of a reproducible model. To our knowledge, these data are the first to test the hypothesis that yeasts may act as potential “keystone” components of oral biofilms.

## 1. Introduction

Biofilms are an integral part of a healthy oral microbiome, with a diverse range of constituents [1]. In dysbiotic states, such as lowered pH or increased plaque accumulation leading to host inflammation, these biofilms can cause a range of diseases, from dental caries, a disease highly prevalent in children, to periodontal disease in adulthood and denture-related infections in the elderly [2]. It has been projected that 90% of the population will need some form of disease intervention in the oral cavity throughout their life, and the World Health Organization defines oral diseases as the most common preventable disease burden worldwide [3,4,5]. This extensive and preventable occurrence of oral diseases highlights the need for further research and accurate clinical treatments for their prevention.

A wide range of bacterial colonizers have been confirmed within a healthy oral cavity through advanced molecular techniques such as next generation sequencing. From such studies, a number of key bacterial genera appear to be conserved amongst individuals, including *Streptococcus*, *Veillonella*, *Granulicatella*, *Actinomyces*, *Fusobacterium* and various anaerobic bacteria [6,7,8]. In addition, *Candida* and other fungal species have also been identified as colonizers in studies focusing on the mycobiome in the oral cavity [9,10,11,12]. Indeed, it is well documented that at mucosal surfaces, *C. albicans* has the ability to act as a symbiont in healthy individuals, transitioning to a pathobiont or dysbiotic entity in certain hosts under some conditions [13,14]. This is similarly the case for the oral cavity [15,16,17], whereby *C. albicans* is commonly associated with known constituents of the oral microbiome in both healthy and diseased individuals [18,19,20,21]. Despite this confirmation, many studies lack a crucial fungal element within in vitro biofilm testing, leading to unrealistic challenges of models designed for the oral cavity [22,23,24]. This lack of interkingdom polymicrobiality limits the advancement and improvement of commercial biofilm testing of novel agents in vitro.

As discussed above, studies have confirmed the presence of numerous fungal species within oral biofilms; however, their role is still disputed. Common commensal plaque constituents, such as *Streptococcus* and *Actinomyces* species, have been shown to grow synergistically with *C. albicans,* increasing biofilm production [25,26]. While this may indicate an increase in pathogenicity when *Candida* is present within plaque, strengthened by known interactions with periodontal pathogen *Porphyomonas gingivalis* (for example, [21]), *C. albicans* has been identified in healthy individuals, suggestive the organism possesses a symbiotic phenotype [9]. Indeed, the ability of *C. albicans* to shift from a “commensal state” as a yeast cell to a “pathogenic” hyphae-forming phenotype under certain conditions also impacts its invasiveness at mucosal sites such as the oral cavity [27,28]. Alternatively, antagonistic interkingdom interactions have also been described whereby pathogens such as *Fusobacterium nucleatum* can inhibit of hyphae formation in *C. albicans*, which leads to reduced virulence attributes of both organisms [29]. From this, it is evident that the interkingdom interactions in the oral cavity can both be beneficial or detrimental to the oral micro- and/or mycobiomes.

Clear evidence suggests that bacteria such as *P. gingivalis* can act as “keystone pathogens” in periodontitis [30]. However, Janus and colleagues (2016) have proposed that *C. albicans* acts as a “keystone commensal” given its close interactions with “health-associated” bacteria, offering a scaffold effect and influence on the local environment within the biofilm through pH buffering [15]. *C. albicans* has been found within and implicated in all aspects of oral disease from caries, periodontitis and denture stomatitis. Close association with *Streptococcus mutans* in caries patients has caused researchers to link *Candida* to a more severe cariogenic lesion in children [31]. Similarly, *Candida* spp. have been heavily implicated in soft tissue infections, such as denture stomatitis and periodontitis with its ability to adhere to and invade epithelial cells [32,33]. These implications may be substantial; however, the presence of *C. albicans* alone does not indicate disease, and so environmental factors are likely significant. The “ecological plaque hypothesis” focuses upon the relationships between the organisms present and the environment within the oral cavity, with disease occurring upon a dysbiotic “event” [34]. Therefore, physiological interventions alongside microbial control may offer beneficial therapeutics [35]. Ultimately, this may indicate that a role for *C. albicans* in health and disease is more closely linked to the interactions and overall environment of the oral cavity, as opposed to fungal burden leading to disease. Thus, including *C. albicans* in in vitro model systems is essential to truly understanding interkingdom relationships within these polymicrobial biofilms.

One aim of this study was to determine whether the inclusion of *C. albicans* within two oral models, a hard-tissue and soft-tissue biofilm, leads to changes in biofilm “profiles” and/or pathogenic “signatures.” Specifically, biomass quantification, metabolic profile, composition and ultrastructure of the biofilms were assessed with or without *C. albicans.* Secondly, an aim was to investigate whether inclusion of this organism will impact the biofilms’ responses to conventional therapeutics. To the authors knowledge, this is the first study to assess the importance of including a fungus in such in vitro systems to model interkingdom oral biofilms and in particular test the “keystone commensal” hypothesis postulated by Janus et al. (2016) previously [15]. Moving forward, results from this study will hopefully inform us of the importance of mimicking fungal–bacterial interactions in vitro for antimicrobial testing, particularly in a commercial setting.

## 2. Materials and Methods

All biofilm studies reported in this work were carried out in accordance with the minimum information guidelines specified for biofilm formation in microplates [36].

### 2.1. Growth and Standardization of Bacteria

The biofilm models used within this study have previously been published by the research group [24,37,38]. These models were created to encompass the well-documented main etiological agents of the oral diseases they are based upon [39,40]. To highlight the natural evolution of disease, sequential addition of the organisms was employed. Initial colonizers were used to attach to the surface of the plate; intermediate colonizers allowed bridging between organisms; pathogenic colonizers were added as a final stage of disease [41].

All pure culture organisms were stored prior to use in Microbank™ beads (Pro-lab Diagnostics, Birkenhead, UK) at −80 °C. Prior to culture preparation, organisms were revived from frozen stocks as follows: *Streptococcus mitis* NCTC 12261, *Streptococcus intermedius* ATCC 27335, *Streptococcus oralis* ATCC 35037, *Streptococcus mutans* ATCC 25,175 and *Aggregatibacter actinomycetemcomitans* DSMZ 1123 were grown and maintained at 37 °C 5% CO_2_ on Colombia blood agar (CBA (Oxoid, Basingstoke, UK)). *C. albicans* 3153A was maintained on Sabouraud’s dextrose agar (Oxoid, UK) at 30 °C under aerobic conditions for 48 h. *Fusobacterium nucleatum* ATCC 1096, *F. nucleatum* ssp. *vincentii* ATCC 49256, *Actinomyces naeslundii* ATCC 19039, *Veillonella dispar* ATCC 27335, *Prevotella intermedia* ATCC 25,611 and *Porphyromonas gingivalis* W83 were maintained at 37 °C on fastidious anaerobic agar (FAA (Lab M, Bury, UK)) in an anaerobic chamber (Don Whitley Scientific Limited, Bingley, UK) with an atmosphere of 85% N_2_, 10% CO_2_ and 5% H_2_ for 48 h or 72 h for strict anaerobes. All media and agar to be used for anaerobes were deoxygenated for 24 h prior to use. *Lactobacillus casei* DSMZ 20,011 was grown and maintained on MRS agar (Oxoid, UK) in 37 °C 5% CO_2_. Aerobic organisms were then stored at 4 °C for a maximum of 2 weeks prior to propagation, while anaerobic organisms were stored within the anaerobic chamber for 1 week prior to being re-streaked.

Culture broths for propagation were prepared as follows: Overnight (16–18 h) broths of *S. mitis*, *S. intermedius*, *S. oralis*, *S. mutans* and *A. actinomycetemcomitans* were grown statically in 10 mL tryptic soy broth (TSB, Sigma-Aldrich, Gillingham, UK) at 37 °C 5% CO_2_. *C. albicans* was grown for 16–18 h in 10 mL yeast peptone dextrose (YPD, Sigma-Aldrich, UK) at 30 °C in an orbital benchtop shaker at 200 rpm, 20 mm orbital diameter (IKA KS 4000 I control, IKA®-Werke GmbH & Co., Staufen, Germany). *P. gingivalis*, *F. nucleatum* and *F. nucleatum* ssp. *vincentii* were propagated in 10 mL deoxygenated Schaedlers anaerobic broth (Oxoid, UK). *V. dispar*, *A. naeslundii*, and *P. intermedia* were grown in 10 mL of brain heart infusion broth, (BHI, Sigma-Aldrich, UK) in anaerobic chamber 37 °C, 85% N_2_, 10% CO_2_ and 5% H_2_. *L. casei* was grown in 10 mL of MRS broth (Oxoid, UK), 37 °C, 5% CO_2_. Anaerobic cultures were grown for 48 h as necessary, then pelleted by centrifugation (VWR megastar 1.6R, VWR International GmbH. Darmstadt, Germany (3000× *g*)). Pellets were then washed via resuspension twice in phosphate buffered saline (PBS, Sigma-Aldrich, UK). Following washing the cells were standardized to 1 × 10^8^ cells/mL using a spectrophotometer for bacteria and via haemocytometer counting for *Candida* (cell count × dilution factor × volume of square = colony forming unit [CFU/mL). Previously, to determining accurate absorbance readings at 550 nm that equate to 1 × 10^8^ cells/mL for the bacteria, the Miles and Misra colony counting technique was employed [42] by serially diluting pure colonies to determine the correct absorbance per organism. *Streptococcus* spp., *V. dispar* and *L. casei* were read at an OD of 0.5. *A. naeslundii*, *F. nucleatum*, *F. nucleatum* ssp. *vincentii*, *P. gingivalis*, *P. intermedia* and *A. actinomycetemcomitans* were read at an OD of 0.2.

### 2.2. Development of Multi-Species Biofilms

Bacteria and fungi were standardized (1 × 10^7^ CFU/mL) in an equal volume of Roswell Park Memorial Institute-1640 (RPMI) with Todd Hewitt Broth (THB) supplemented with 0.01 mg/mL hemin and 0.2 µL/mL (2 µg/mL) menadione in a similar manner as previously described [43]. These preparations were generated by diluting bacteria 10× after the initial standardization following washing and diluting to appropriate OD. *Candida* dilution was carried out using the CFU/mL calculated previously and applying the C1V1 = C2V2 calculation. Biofilms formed to create the oral biofilm models were grown directly in a clear flat bottomed 96 well polystyrene plate (Corning, New York, US) using a 200 µL inoculation volume unless stated otherwise. Medium was replaced every 24 h using a multichannel pipette, carefully at a 45° angle and at the same point within the plate (bottom left of the well) to limit unnecessary disruption of the biofilm.

### 2.3. Hard Tissue (HT)—Caries Associated Biofilm

Initial pioneer species *C. albicans* and *S. mutans* were standardized together within an RPMI/THB aliquot and matured in 96 well flat bottom plates using 200 µL per well for 24 h in 5% CO_2_. The supernatant was then removed as described above and an RPMI/THB aliquot containing a mixture of standardized *F. nucleatum*, *A. naeslundii*, *V. dispar* and *L. casei* was added and incubated in the same conditions for 4 days. RPMI/THB medium was replaced daily to produce a 6-species biofilm. The supernatant was then removed, and biofilms were ready to be used for downstream experiments. In biofilms omitting *Candida*, the same sequential addition was used minus *C. albicans* in the initial step.

### 2.4. Soft Tissue (ST)—Periodontitis/Denture Stomatitis Associated Biofilm

Initial pioneer species *C. albicans*, *S. oralis, S. mitis* and *S. intermedius* were standardized together in an RPMI/THB aliquot and matured in 96 well flat bottom plates using 200 µL per well for 24 h in 5% CO_2_ The supernatant was removed as described above and an RPMI/THB aliquot containing a mixture of standardized *F. nucleatum*, *F. nucleatum* ssp *vincentii*, *A. naeslundii* and *V. dispar* was added and incubated in an anaerobic cabinet (85% N_2_, 10% CO_2_, 5% H_2,_ (Don Whitley Scientific Limited, UK)) for 24 h. The supernatant was again removed and a deoxygenated RPMI/THB aliquot containing standardized *P. gingivalis*, *P. intermedia*, and *A. actinomycetemcomitans* added in an anaerobic cabinet. The medium was replaced daily for 4 days to produce an 11 species biofilm. The supernatant was then removed, and biofilms were ready to be used for downstream experiments. As above, in biofilms omitting *Candida*, the same sequential addition was used minus *C. albicans* in the initial step.

### 2.5. Testing of Oral Models

To establish the importance of incorporating *C. albicans* into the biofilms, assessment of metabolic testing alongside biomass determination was carried out. Additionally, qPCR was utilized to assess changes in bacterial load and composition. Scanning electron microscopy was also used to determine architectural differences. Changes in pathogenesis were also estimated by measuring fluctuations in pH, reactive oxygen species (ROS) and enzyme levels. To further define the differences in the biofilms with or without *C. albicans,* models were subjected to treatment as described below.

### 2.6. Metabolic Analysis of Oral Biofilms

Biofilms were analyzed for metabolic activity using alamarBlue™ cell viability dye (Invitrogen, Inchinnan, UK). Once matured or after treatment and neutralization, alamarBlue™ was added at 10% well volume in RPMI/THB (e.g., 20 µL per 200 µL RPMI/THB) using a multichannel pipette in dark conditions. Plates were then incubated at 37 °C 5% CO_2_, with a change from blue to pink indicative of a reduction in the fluorogenic dye resazurin to resorufin by enzymes involved in normal cellular respiration. The color change was read when the growth control reached a bright pink color to allow this to be used as a positive control or the latest after 3 h, depending on which occurred first. 100 µL from each well was transferred to a fresh 96 well flat bottom microtiter plate and top read using fluorescence 530/590 nm (FLOUstar™, BMG lab tech, Aylesbury, UK). A negative sterility control was also used on the same plate whereby no biofilm growth was initiated to use as a blank to normalize the fluorescence by an average of this value. In cases of treatment, a positive growth control (e.g., minus treatment) was included as a 100% growth value, to allow metabolic activity to be calculated as a percentage based on this growth control. All experiments assessing metabolic activity were completed using biofilms in triplicate or quadruplicate on three separate occasions.

### 2.7. Assessment of Oral Biofilms Biomass

The biomass of the biofilms were quantified using a crystal violet stain. Post metabolic assessment, spent cell viability dye was removed carefully with the pipette tip placed at the bottom left of the well, then biofilms were washed once with PBS and airdried overnight at RT. A stock solution of 1% crystal violet (CV) (Sigma-Aldrich, UK) was made using ddH_2_O and diluted to 0.05% for use. A total of 100 µL of 0.05% CV solution was added to each well and incubated at RT for 30 min. CV was then removed using a multichannel pipette, and biofilms washed twice using ddH_2_O removing excess dye. To limit unnecessary disruption of the biofilm, the wash steps were achieved by placing the pipette tip at 45º angle at the same point within the plate (bottom left of the well). Following wash steps, a volume of 100 µL of 100% ethanol was used to de-stain the biofilm; following addition to the biofilms (at the same point within the plate as above e.g., bottom left of the well), this suspension was mixed well with a pipette five times and 75 µL transferred to a fresh 96 well flat bottom microtiter plate for measurement. Biomass was quantified by reading absorbance at 570 nm (FLOUstar™, BMG lab tech, UK). A negative sterility control was also used on the same plate whereby no biofilm growth was initiated to use as a blank to normalize the absorbance by an average of this value. In cases of treatment, a positive growth control was included, to be used as an untreated comparison. All experiments assessing biomass were completed using biofilms in triplicate or quadruplicate on three separate occasions.

### 2.8. Quantitative Analysis of Biofilm Composition

Real-time quantitative PCR (qPCR) was performed to enumerate the relative composition of the biofilms. Briefly, bacterial biofilms were removed by using a pipette tip in 150 µL PBS to detach the cells from the 96-well plate and added to 850 µL PBS for further use. Initially, following removal of these biofilms using the pipette, a plate was stained with CV to ensure no biomass remained. Indeed, using this methodology, minimal biomass remained on the plate after removal. The homogenized suspension was then used for DNA extraction using the QIAamp DNA Mini Kit (Qiagen, Manchester, UK), following manufacturer’s instructions. Following on from this, 1 μL of extracted DNA was added to a mastermix containing 10 μL SYBR^®^ GreenER™, 7 μL UV-treated RNase-free water and 1 μL of 10 μM forward/reverse primers for each bacterial genus or species. The primers used were previously published, as listed elsewhere [24,38], and are shown in Table 1. Three independent replicates from each parameter were analysed in duplicate using StepOnePlus Real-Time PCR system and StepOnePlus software (ThermoFisher, Loughborough, UK). PCR thermal profiles were as follows, holding stage at 50 °C for 2 min, followed by denaturation stage at 95 °C for 5 min and then 40 cycles of 95 °C for 3 s and 60 °C for 30 s. Samples were quantified to calculate the colony forming equivalent (CFE) based upon a previously established standard curve of bacterial colony forming units ranging from 1 × 10^3^ to 10^8^ CFU/mL. The R_2_ values for these standard curves ranged from 0.984 to 0.997. All samples from each triplicate experiment were run in duplicate within the qPCR plate and used as a mean to limit fluorescence variation. Two duplicate control samples containing mastermix only were used to assess for contamination.

### 2.9. Visualization of Oral Biofilms

Scanning electron microscopy (SEM) was performed on intact biofilms grown on 13mm^2^ coverslips (Thermofisher, UK). Following maturation and treatment, biofilms were washed three times with PBS with the pipette tip placed in the bottom left of the well to avoid disruption of the biofilm. Following washing, fixative solution was then added to the biofilms containing 2% paraformaldehyde, 2% glutaraldehyde, 0.15 M sodium cacodylate and 0.15% *w/v* alcian blue (pH 7.4) and fixed overnight. A 0.15 M sodium cacodylate buffer was then used as a storage buffer and samples kept at 4 °C until processing. SEM processing of biofilm samples followed a series of fixation and dehydration steps preceding a final gold sputter coating allowing visualization as described elsewhere [52]. Briefly, samples were washed 3 × 5 min with 0.15 M sodium cacodylate to ensure glutaraldehyde removal. Samples were then treated with 1% osmium tetroxide solution containing 0.15 M sodium cacodylate (1:1) and incubated in a fume hood for 1 h. Samples were rinsed 3 × 10 min with distilled water and then treated with 0.5% uranyl acetate and incubated in the dark for 1 h. Uranyl acetate was removed from the samples and rinsed with water before a series of dehydration steps were carried out. 2 × 5 min rinses of 30, 50, 70 and 90% alcohol were followed by 4 × 10 min rinses of absolute and dried absolute alcohol. Critical point drying was then undertaken by soaking the biofilms in hexamethyldisilazane for 2 × 5 min rinses. Samples were then placed in a desiccator overnight to allow samples to be thoroughly dried. The specimens were then mounted and sputter-coated with gold in an argon filled chamber, and then viewed under a JEOL JSM-6400 scanning electron microscope.

### 2.10. Assays to Assess Pathogenicity of the Oral Biofilms

Supernatants were collected from the biofilms on the final day of growth, centrifuged in microcentrifuge tubes at 8000× *g* for 5 min. Aliquots of the supernatant were used fresh for pH testing (FiveEasy meter and InLab^®^ micro (Mettler Toledo, Columbus, OH, USA)) and the remainder frozen at −80 °C until further use for remaining assays. Biofilms were tested using lactate assay kit as per kit instructions (Sigma-Aldrich, UK). Briefly, 20 µL of sample was mixed with a master mix (1:1) consisting of assay buffer, enzyme mix and lactate probe in a 23:1:1 ratio. A standard curve was also prepared. Mixture was incubated in the dark for 15 min at room temperature and read at 570 nm (FLOUstar™, BMG Labtech, UK). ROS assessment was carried out using a superoxide determination kit as per manufacturers instruction, with the exception of incubation time (Sigma-Aldrich, UK). Briefly, 20 µL of sample was added to a 96-well microtiter plate and mixed with 200 µL of diluted WST working solution and 20 µL of diluted enzyme working solution. This was then incubated at 37 °C for 24 h and absorbance read at 450 nm (FLOUstar™, BMG Labtech, UK). Appropriate blanks were used according to instruction of the kit. Each replicate sample were run in duplicate. Superoxide dismutase enzyme from *Escherichia coli* (Sigma-Aldrich, UK) was used to achieve an inhibition standard curve to allow units/mL to be calculated from percentage inhibition. Protease release was determined using a protease fluorescent detection kit following manufacturer’s instructions (Sigma-Aldrich, UK). In a microcentrifuge tube, 10 µL of each sample was used neat and mixed with 20 µL of incubation buffer and 20 µL of fluorescent labelled casein and incubated for 1 h at 37 °C in the dark. A total of 150 µL of 0.6N Trichloroacetic acid solution was then added and mixed to precipitate larger fragments which have not been cleaved. After incubation at 37 °C for 30 min in the dark, the tubes were centrifuged for 10 min at 3000× *g*. The supernatant was transferred to a black 96-well microtiter plate and read as a fluorescent measurement at 485/535 nm, excitation/emission (FLOUstar™, BMG Labtech, UK). Serine protease trypsin was used as a control to generate a standard curve and negative controls used as appropriate.

### 2.11. Treatment of Oral Biofilms

To determine if the inclusion of *C. albicans* affected the susceptibility or tolerance of the oral biofilms to treatment, a range of both therapeutic challenges were used. The following concentrations and treatment preparations were used; 0.2% chlorhexidine gluconate (CHX, (Sigma-Aldrich, UK)) was used to mimic standard daily mouth-washing as a gold standard used within the dentistry industry and was used to mimic standard daily mouth washing [53], 4% Ethylenediaminetetraacetic acid (EDTA, (Sigma-Aldrich, UK)) a commonly used irrigant in endodontic treatments was used [54]. Potassium iodine at 1% served to mimic disinfection protocols in dentistry procedures [55]. Miconazole, an ergosterol inhibitor and anti-fungal agent was used at a concentration of 40 µg/mL based on previous studies [56]. All solutions were prepared fresh in sterile H_2_O prior to experimental use.

For the treatment studies, biofilms were grown in 96-well microtiter plates then washed three times to remove non-adherent cells as above. A total of 200 µL of testing compound was pipetted onto the biofilms at a 45° at the side of the well to avoid any disturbance to the biofilm. Biofilms were incubated at room temperature for 10 min. This incubation time was selected so that it partially mimicked the short treatment for oral health products, but long enough to ensure efficient action of the therapeutics, especially since this model utilized a static system (e.g., without combinative physical debridement). Following incubation, treatments were neutralized using 200 µL Dey Engley neutralizing broth (Sigma-Aldrich, UK) for 15 min and then washed once more with PBS using the same procedure as discussed above. A total of four wells within the microtiter plate were then used for metabolic analysis and biomass determination per experiment, while three others were used solely for biomass removal and DNA extraction for qPCR analyses. All experiments assessing metabolic activity were completed using biofilms in triplicate or quadruplicate on three separate occasions.

### 2.12. Statistical Analysis and Data Presentation

Graph production, data distribution and statistical analysis were performed using GraphPad Prism (version 8; La Jolla, CA, USA). After assessing whether data conformed to a normal distribution, unpaired Mann–Whitney *t* tests were used to investigate significant differences between independent groups of data. A Kruskal-Wallis was carried out with Dunn’s post-test analysis to assess differences between biofilms under treatment. Statistical significance was achieved upon *p < 0.05*.

## 3. Results

### 3.1. Metabolic, Biomass and Compositional Profiles of Oral Biofilm Models with or without C. albicans

Multi-species biofilms were grown in parallel in the presence or absence of *C. albicans* to determine the change to the overall metabolic rate and biomass of biofilms following incorporation of *Candida* (Figure 1). The results indicate that both metabolic activity and biomass were significantly increased upon the inclusion of *C. albicans* in both hard (HT) and soft tissue (ST) biofilms (*p* < 0.001). The fluorescence readings illustrating metabolic activity showed that HT biofilms increased by 2.8-fold, the mean output reading went from ~16,000 to ~45,000 Au (Figure 1A). Meanwhile, ST biofilms showed a 1.8-fold increase in metabolic activity when *C*. *albicans* was present, increasing from ~27,000 to ~50,000 Au. In a similar trend, the biomasses of the oral biofilm models were also significantly increased when *Candida* was included; HT biofilms increased almost 4-fold from an average O.D. of ~0.5 to ~2.0; and ST associated biofilm increased from ~0.8 O.D. to ~2.0, a 2.5-fold increase (Figure 1B). Overall, this data highlighted a thicker biofilm which is more metabolically active when *C. albicans* was present, in both models tested.

Crystal violet staining is unable to differentiate between cell numbers and composition and is just an overall estimate of the biomass of all components (cells, eDNA and extracellular matrix). Therefore, quantitative PCR was used to determine bacterial and fungal load using 16S and 18S primers (Figure 2A). Additionally, genus and species-specific primers detailed in Table 1 were also used to calculate overall percentage composition in HT (Figure 2B) and ST biofilms (Figure 2C). Results show that bacterial load was increased in both HT and ST biofilm models when *C. albicans* was present (both *p* < 0.001). A significant increase of more than one log was seen in the HT biofilm, ~6.0 × 10^7^ CFE/mL, which increased to ~1.5 × 10^9^ CFE/mL when *C. albicans* was included. Similarly, an increase from ~1.8 × 10^8^ to ~9.0 × 10^8^ CFE/mL was observed in ST-associated biofilms. The fungal load present in each of the biofilms was also assessed. Results showed that ~4.4 × 10^7^ CFE/mL and ~6.2 × 10^6^ CFE/mL of *C. albicans* were detected in HT and ST, respectively.

Following on from total bacterial load assessment, compositional analysis of the biofilms was done. In the HT biofilm, Figure 2B, the key change associated with the inclusion of *C. albicans* was the reduction of *S. mutans* from ~43.1% in biofilms without *C. albicans*, to ~29.0% in biofilms with *C. albicans*. However, this compositional change was not due to a reduction in microbial burden of *S. mutans,* which increased from 7.12 × 10^6^ to 1.04 × 10^8^ CFE/mL. *A. naeslundii* and *L. casei* also reduced in proportion, following inclusion of *C. albicans*, decreasing from ~7.8% to ~1.2%, and ~0.01% to ~0.001%, respectively. Again, this reduction was not associated with reduced numbers of this organisms (1.29 × 10^6^ compared to 4.25 × 10^6^ with *C. albicans* for *A. naeslundii* and 1.99 × 10^3^ vs. 4.19 × 10^3^ plus *C. albicans* for *L. casei*). All mean CFE/mL and compositional changes are documented in Table 2.

Within the ST biofilm (Figure 2C), all bacterial genus or species increased in CFE/mL in the presence of *C. albicans*. Compositionally, *V. dispar* and *Streptococci* species (*S. mitis*, *S. oralis* and *S. intermedius*) remained the main constituents of the biofilm after the inclusion of *C. albicans*, although a reduction in the proportion of *Streptococcus* species was seen (~79.3% compared to ~89.9%, for *V. dispar*, and ~14.40% compared to ~3.26%, for *Streptococci*, respectively). Meanwhile, *A. naeslundii* (~4.76% to ~1.55%) and *A. actinomycetemcomitans* (~1.56% to ~1.25%) were both reduced in proportion following addition of *C. albicans*. The Socransky red complex constituent of *P. gingivalis* was present at low quantities, as expected in a periodontal biofilm [40]. with the organism increasing ten-fold in proportion and 100-fold in concentration after the addition of *C. albicans* (0.001% and 1.23 × 10^2^ CFE/mL in bacterial-only biofilms, compared to 0.011% and 1.74 × 10^4^ CFE/mL for *Candida*-containing biofilms, respectively). The bridging organism *F. nucleatum* also showed an increase from ~0.01% to ~0.05% in the presence of *C. albicans*, including a ~2-fold increase in bacterial numbers (2.80 × 10^3^ compared to 5.12 × 10^3^ CFE/mL). All CFE/mL and % changes are shown in Table 2. Overall, the results indicated that addition of *C. albicans* to the biofilm increased the bacterial load in both oral biofilm models. Changes associated with the composition showed a more diverse biofilm in HT biofilm, whilst ST biofilms were still dominated by *V. dispar* and *Streptococcus* spp. An increase in the periodontal pathogens such as *F. nucleatum* and *P. gingivalis* in the ST biofilms indicate a more diverse consortium with the inclusion of *C. albicans.*

### 3.2. Scanning Electron Microscopy Visualization of Oral Biofilm Models with or without C. albicans

Next, the ultrastructures of the oral biofilm models were assessed using scanning electron microscopy (SEM). SEM imaging highlighted the change in architecture, particularly in HT biofilms wherein *C. albicans* was included (Figure 3). In HT biofilms without *C. albicans* (Figure 3A), the cells were clustered into complexes that were spaced out across the coverslip, in stark contrast to the network of cells and ECM within HT biofilms, including a fungal element (Figure 3B). ST biofilms (Figure 3C) demonstrated a complex and dense biofilm without the addition of *C. albicans*. Although it is clear from the bacterial load that there was still a small increase in bacterial numbers (as shown in Figure 2A), SEM pictures were comparable between the biofilms plus or minus *C. albicans* for the ST biofilms (Figure 3C,D). Nevertheless, there were clear physical interactions between the microbiota and the yeast and/or hyphae of the *C. albicans* in both models (indicated by the red arrows in the insets of Figure 3B,D).

### 3.3. Change in Pathogenic Biomarkers in Oral Biofilm Models in the Presence and Absence of C. albicans

Next, a range of biomarkers for virulence were assessed to determine if fungal addition to the oral biofilm models altered the pathogenicity of the biofilms. The pHs of the oral cavity and saliva have been established as markers for disease [57,58], whilst ROS and protease production may be linked to toxic metabolites and virulence [59,60]. Therefore, these markers were assessed in the two biofilm models (Figure 4). The pHs of both biofilm supernatants were significantly increased to a more neutral value in the presence of *C. albicans* (Figure 4A). HT biofilm supernatants had an average pH value of 5.8, which increased to 7.4 and ST biofilm supernatants increased by 1.5 units to a pH of 6.5 following inclusion of *C. albicans*. This was suggestive that the presence of a fungal element acts as a buffer to increase the environmental pH, which is in line with previous observations for *C. albicans* [61]. To corroborate these observations in pH measurements, lactate concentration was also determined. The modulation of lactate production by the microbial consortia is a common mechanism to generate acidic environments, which is likely occurring in the cariogenic HT biofilm. Results from Figure 4B demonstrate that in both HT and ST biofilms, *C. albicans* reduced lactate content in the supernatant collected from the biofilms. A 2-fold reduction was noted from HT biofilms, while a 14-fold decrease was apparent in ST biofilms following addition of *C. albicans*.

ROS within the oral cavity can be a marker for inflammation and cell damage [62], and therefore the activity of enzyme superoxide dismutase was determined in the biofilm supernatants. Interestingly, in HT biofilms, *Candida* decreased superoxide dismutase activity 5-fold, whilst conversely, the presence of *C. albicans* increased the activity in the ST biofilms 4-fold (Figure 4C). Finally, similarly to ROS, protease activity can be an indicator of a dysbiotic biofilm [63]. Figure 4D shows an increase in protease activity in HT biofilms when *C. albicans* is present, with a 9-fold increase in protease activity. In contrast, quantities remained low in both ST biofilms with a small 1.6-fold increase in those biofilms with *C. albicans*.

Figure 5, panels A and C, showed that the most effective treatment was CHX, which reduced the metabolic activity of both biofilms tested, irrespective of the inclusion of *Candida*. The metabolic activities of the remaining biofilms were comparable to untreated controls following treatment with EDTA, potassium iodine (KI) or miconazole (MCZ), even following the addition of *C. albicans*. Post-treatment with CHX, metabolic activity was reduced by about 50% compared to the control in HT (Figure 5A) and ~70% in the ST biofilms (Figure 5C) regardless of inclusion of a fungal element, which was a significant reduction when compared to each individual control (*p* < 0.01, *p* < 0.001). In contrast, average metabolic activity in the biofilms following treatments from EDTA, KI and MCZ showed small changes from the control, but not significantly different. For the CHX-treated ST biofilm minus *C. albicans*, significantly greater biomass was detected when compared to the controls, suggestive of possible artificial CV staining of CHX precipitate within these biofilms (*p* < 0.001). There were no significant differences in susceptibility profiles of the HT biofilms to the treatments compared to the untreated controls (Figure 5B). Interestingly, ST biofilms treated with EDTA were the only treated biofilm to become more susceptible to disruption compared to the UT following the inclusion of *C. albicans* (Figure 5D; # *p* < 0.05), while all other treatments remained comparable to the control. Overall, this highlights a similarity in the treatment responses of the biofilms, which following the inclusion of *C. albicans,* did not create a more resilient or susceptible biofilm in vitro.

### 3.4. Bacterial Load Following Treatment in Oral Biofilm Models with or without C. albicans

Finally, the bioburden of bacteria and fungi were assessed following treatment using qPCR analyses. For all treatments, no significant difference was found when comparing bacterial load (16S) to the respective untreated controls in either biofilm model (Figure 6A,C, white bars). Similar results for bacterial counts were observed when *C. albicans* was included in the models (ns, *p* > 0.05; Figure 6A,C, grey bars). In both biofilm models, irrespective of treatment, log increases of between ~0.5 and ~1.5 units were seen in bacterial counts following inclusion of *C. albicans* (Figure 6B,D). No differences were also seen when comparing the fungal counts (18S) following treatment, with *C. albicans* maintaining a level of ~3–5 × 10^7^ CFE/mL for HT, and ~3–6 × 10^7^ CFE/mL for ST irrespective of treatment type (ns, *p* > 0.05). All average CFE/mL values for each biofilm with or without treatment are shown in Table 3. Taken together, the qPCR data indicated that the bacterial load was not affected by treatment with a variety of different agents following incorporation of *C. albicans*, suggesting that the fungal element did not convey protection for the microbiota against the variety of agents.

## 4. Discussion

*Candida albicans* is known as an opportunistic pathogen which colonizes many different sites of the human body, with an ability to cause severe disease through candidiasis in the immunocompromized [64]. *C. albicans* also has the ability to form biofilms on a range of surfaces in a medical setting, such as indwelling medical devices [65], and has also been found as a potent colonizer of various oral mucosal sites [66,67]. With known mortality rates of up to 60% in candidemia patients and studies highlighting the pathogenic nature of *Candida* biofilms [68,69], from a clinical standpoint, *Candida* infections and associated biofilms are viewed as highly problematic. Furthermore, in the context of biofilm research, the pathogenic mechanisms of many microbial diseases are now often viewed as interkingdom and polymicrobial in nature [70,71]. As such, it is perhaps no surprise that our understanding of fungal–bacterial interactions continues to progress, particularly in the oral cavity whereby *C. albicans* specifically has been implemented as potential “active accomplice” in different oral infections [11]. Hence this study aimed to investigate the biofilm “profiles” and/or pathogenic “signatures” of two oral biofilm models following incorporation of *C. albicans.*

Firstly, and perhaps unsurprisingly, results showed that metabolic activity and biomass were increased when *C. albicans* was included within the biofilms (Figure 1). Metabolic activity quantified via Alamar blue™ assay is a measurement of the rate of respiration in the reduction of resazurin to resorufin by electron acceptors such as NADPH and cytochromes [72]. Crystal violet staining, commonly used in Gram staining techniques, binds to the cell walls of bacteria and fungi, and lipopolysaccharides and DNA, both of which are large components of the extracellular matrix (ECM) [73]. Metabolic activity and biomass testing in biofilms in reality are crude measurements that can be useful when used with a variety of other techniques. Hence, it is essential to note that growth rates differ between organisms, and therefore metabolic output observations should be viewed with a certain level of caution. As such, metabolic activity may not be representative of the whole biofilm, but as an indicator of the fastest growing and thereby more metabolically active organism present in the consortia. However, it can still be used as an inclusive measurement tool with other techniques to assess changes in the biofilm environment [74,75]. Additionally, CV staining does not differentiate between additional components of the biofilm (e.g., extracellular matrix and eDNA) and cells within the biofilm [73,76]. Nevertheless, results from these simple methodologies highlight that a thicker, more metabolically active biofilm was formed in the presence of *C. albicans*.

From the results here, the increase in metabolic activity and biomass for biofilms grown with *C. albicans* could be multifactorial and not simply caused by more cells being present. Therefore, it was deemed pertinent to assess microbial cell numbers via qPCR. For the CV observations, the size difference of *C. albicans* to bacteria may influence the increase in CV values, with the average *C. albicans* cell being almost 150× larger than their bacterial counterparts, and so it would be logical that an increase in CV was detected [15,17]. Alternatively, the elevated biomass may be caused by an increase in ECM production from cells; studies have shown an increase in ECM production from bacteria in the presence of *C. albicans* [77]. As a culmination of factors, the addition of *C. albicans* may also reduce the amount of biofilm washed away during the CV experimental procedure, helping one to retain a more robust and metabolically active biofilm in vitro. Indeed, *C. albicans* has previously been shown to be a structural scaffold within dual-species biofilms [56]. This may be an effect we observed in this study, particularly with the increase in bacterial load, as shown in Figure 2, and confirmed through increased architecture in SEM images shown in Figure 3, particularly in the HT biofilms. Ultimately, it could be that a robust biofilm containing *C. albicans* would provide a better representation of the polymicrobial nature of the biofilms in the oral cavity, which may allow for more appropriate and accurate routine testing to be undertaken in vitro. At this juncture, it has been shown previously that mechanical removal through brushing alongside therapeutic activity is required to successfully manage these fungal–bacterial biofilms in vitro [38].

Alongside the increase in bacterial load there were compositional changes that accompanied the inclusion of *C. albicans,* as seen in Figure 2B,C. In HT biofilms, the main components remain similar; however, *S. mutans* and *L. casei* decreased in proportion after the addition of *C. albicans*, which may be a result of the increase in pH to a more neutral environment in the presence of the fungi (Figure 4). It has been shown elsewhere that *C. albicans* has the ability to neutralize an acidic environment, a process that will lead to induction of the yeast-hyphal transition, an important virulence mechanism used by the organism to promote survival and/or tissue invasion [61]. The presence of the remaining organisms within this model, even following incorporation of *C. albicans*, was deemed essential to include to provide a representative caries biofilm rather than focusing on *S. mutans* as a single species, a limitation of previous studies given the polymicrobial nature of the disease [78]. In ST biofilms, an increase in pathogens such as *P. gingivalis* was noted following addition of the fungi; therefore, in a similar manner to the HT model, it would allow a more appropriate biofilm to be used to study soft tissue diseases such as denture stomatitis and periodontitis. As such the ST biofilm incorporates anaerobic bacteria representing periodontitis and *C. albicans* in denture stomatitis alongside an interkingdom biofilm representative of that seen in ecological niche of these diseases [19]. *C. albicans* has been shown to generate hypoxic conditions in a biofilm microenvironment [79], which may explain the elevated number of anaerobic pathogens, such as *P. gingivalis* and *F. nucleatum* in the ST biofilms containing *Candida*. Overall, it could be argued that inclusion of any fungal element into such a polymicrobial biofilm would cause composition changes, simply due to the size and morphological phenotypes of such organisms. This, of course, raises the question: “can all oral fungal species serve as keystone commensals?” Such a postulation goes far beyond the remit of the current paper but definitely requires careful consideration moving forward.

The link between acidic conditions in the oral cavity and dental caries onset is well documented; however, periodontal biofilms have also been shown to acidic in nature [80]. As discussed above, *C. albicans* inclusion seemed to offer a buffering effect, as shown in the pH increase with the value closer to neutral in the supernatants collected (Figure 4). Such a buffering effect is not unheard of with *C. albicans*, an organism which prefers to maintain a cytosolic pH range of between 5.8 and 9 [81]. Lactate and lactic acid are common metabolites of oral bacteria, and often a dysbiosis in the utilization and production of these compounds can lead to disease [82]. Results showed here that lactate was reduced in biofilms with *C. albicans* (Figure 4B). *C. albicans* is able to utilize lactate as a substrate to produce an increased stress response and drug resistant properties [83]. Previous studies have also shown that through ammonia production, *C. albicans* cells are able to self-regulate pH to allow morphogenesis (the ability to shift from yeast to hyphae) and proliferation [61]. This rise in pH, likely through the regulation of lactate levels, may suggest that having *C. albicans* within an oral biofilm may provide a protective, more pH neutral environment within this model. Such observations may have implications on studies interested in health vs. disease models, whereby inclusion of *C. albicans* stabilizes the biofilm, making it more habitable by health-associated organisms.

ROS such as superoxide anion (SOD) and hydrogen peroxide are essential metabolites in cellular respiration; however, accumulation and failure to neutralize ROS leads to oxidative stress and cell death [84]. In that capacity, SOD was tested for in the oral biofilm models. Figure 4C shows that SOD is reduced in HT + *C. albicans* containing biofilms, possibly indicating less reactive oxygen species are present within the environment. On the contrary, ST biofilm + *C. albicans* showed an increase in SOD; this may have been due to the facultative anaerobic bacteria (such as *F. nucleatum*, *V. dispar*, *A. naeslundii* and *A. actinomycetemcomitans*) producing more SOD than aerotolerant species [85].

To assess overall hydrolytic qualities of the biofilm models, protease activity was quantified (Figure 4D). *C. albicans* containing HT biofilm showed increased levels of protease activity when compared to the *C. albicans* lacking biofilm. *C. albicans* pathogenicity depends on the secretion of proteinases to invade host cells through the damaging of cell membranes, and so this increase may be directly linked to *C. albicans’* presence [86,87]. In order to fully assess the implication of this, additional work carried out under a high sugar environment would be required. Furthermore, these observations regarding protease activity may simply be due to the increased microbial load within the biofilms containing *C. albicans*. To investigate whether this was the case, experiments presented here would need to be repeated with (1) heat-inactivated *Candida* or (2) mutated isolates lacking specific protease genes (e.g., secreted aspartyl proteinase mutants).

Finally, to assess whether the inclusion of *C. albicans* affected the biofilms response to treatments, the efficacies of different antimicrobials were tested. The antimicrobials chosen represent a series of agents with different mode of actions and purposes, a gold-standard mouthwash (CHX), irrigants (EDTA/KI) and an antifungal agent (MCZ). Following treatment, the metabolic activity, biomass and fungal/bacterial load of each biofilm were assessed. Such parameters were chosen for analyses as previous evidence has indicated that biofilm viability and metabolic activity are reduced following short-term treatment with certain therapeutics (e.g., CHX) [88,89,90], whilst others have shown components such as EDTA can destabilize and degrade extracellular matrix [91], possibly leading to detachment of the biofilm and loss of overall bioburden. From the actives tested, CHX was the only agent which significantly reduced metabolic activity in comparison with the UT control (Figure 5A,C), although this reduction did not equate to detachment since biomass data and bacterial load data are not decreased and similar to the UT control (Figure 5B,D). Furthermore, the efficacy of CHX did not change in biofilms with or without *C. albicans*. EDTA significantly reduced the biomass of the *Candida*-containing biofilm, suggestive that the antimicrobial is effective against the fungal element, which is in line with previous observations [92,93]. The lack of reduction in metabolic activity, biomass and/or cell number for any of the other treatments tested highlights the importance of treatment time and the potential adjunctive use of mechanical disruption alongside antimicrobial therapies, even in the context of in vitro studies. Standard overnight incubations usually employed in minimum biofilm eradication studies may overestimate the effectiveness of the active [94] and so a short incubation time was favored for this study. It may be that longer incubation times would result in differences in tolerance levels of the biofilms to such therapeutics following inclusion of *C. albicans*. It should also be noted that only normal qPCR, and not live/dead qPCR, was used to quantify the bacterial and fungal load in the models following treatment. This live/dead methodology has been used with great effect to assess the level of “viability” of oral microorganisms in plaque samples and in vitro biofilm models [38,44,95]. Therefore, it could be that no changes were observed in the total bacterial or fungal load in this study, but there may have been reductions in the number of viable cells present in the biofilm (which could corroborate with the metabolic activity changes particularly for CHX treatment for example). It is not uncommon for dead microbial cells to remain in the biofilm even following treatment [88,89,96]. Therefore, this poses the question: “does the biofilm require some degree of mechanical disruption for sufficient removal of dead and viable cells?” Indeed, in the case of most oral diseases of microbial origin, particularly for dental caries and periodontitis, the primary aim for clinicians is physical debridement of the biofilm, before administering antimicrobials to prevent the regrowth of microorganisms. This ultimately highlights the need for mechanical removal of biofilms in vitro to truly provide realistic model systems as shown with previous studies conducted [38,97]. Sherry et al. (2016) highlighted the importance of adjunctive therapies of biofilm testing in vitro. In this study, the 11- species biofilm used here (ST) was subject to cleaning regimes involving a denture cleansing treatment. Mechanical disruption by brushing alone was not sufficient to reduce bioburden on the denture substrates, whereby only when used in conjunction with the denture cleanser was significant reductions in viable microbial cells observed [38]. Future studies must consider using these “disrupt and inhibit” experiments with such in vitro oral biofilm model systems.

From this work it can be concluded that future studies should consider incorporation of a fungal element to fully encompass the polymicrobial nature of the oral cavity. The emerging postulations that *C. albicans* may serve as a “keystone commensal” in multispecies oral communities highlights its importance in the context of oral health (and disease) [15]. This study, to the authors knowledge, was the first to truly assess whether *C. albicans* serves as a “keystone” component in oral biofilm models in vitro. From the data, the importance of including *Candida* in such biofilm models is attributed to a more robust structure of the mixed species consortia, while not affecting the tolerance of the biofilm to short-term exposure of conventional oral therapeutics. Indeed, further studies are required to truly understand what role *C. albicans*, and other fungal inhabitants, play in homeostasis of the oral cavity, and we hope this study will provide a platform for further research into said phenomenon. In the long term, inclusion of such an organism in otherwise bacterial-only biofilms should hopefully generate a more applicable model system for testing conventional and/or novel therapeutics.

## Figures and Tables

**Figure 1 microorganisms-09-00059-f001:**
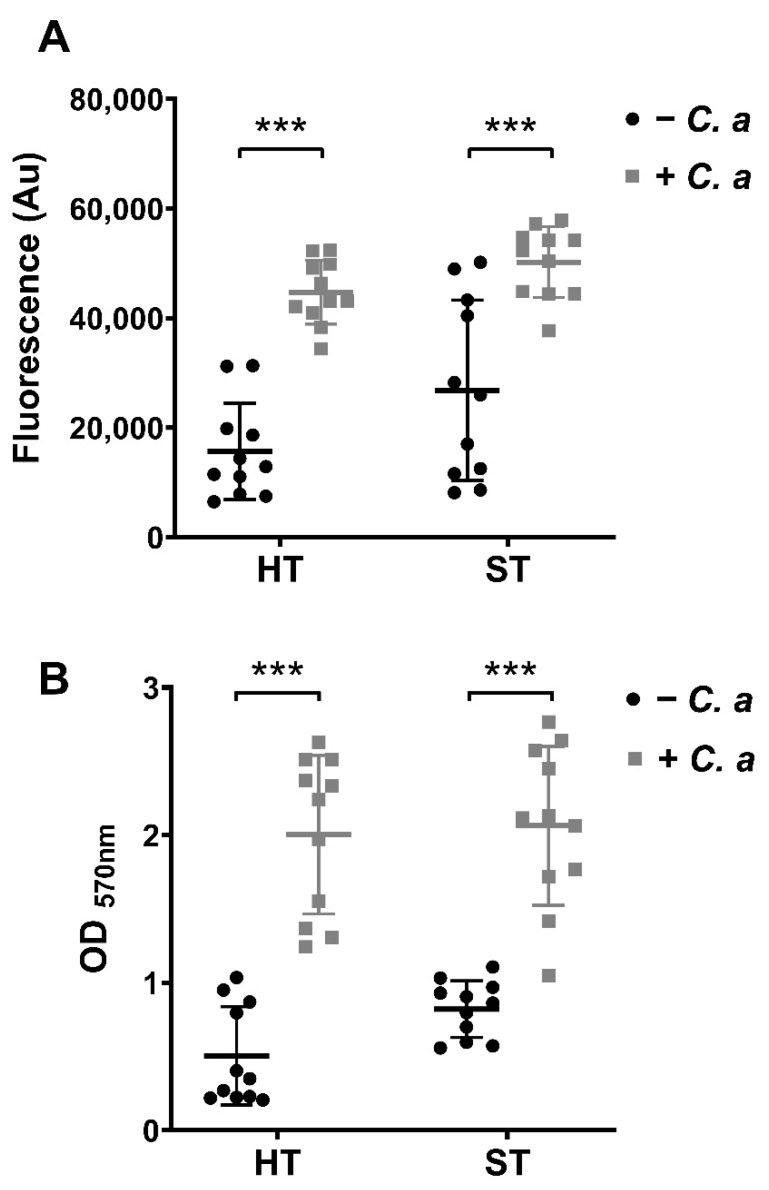
Metabolic and biomass profiles of hard tissue (HT) and soft tissue (ST) biofilms plus or minus *Candida albicans*. Biofilms were grown in 96 well microtiter plates. Metabolic activity (**A**) was assessed using Alamar blue™ rezasurin dye to determine respiration rates, and biomass (**B**) was assessed using crystal violet as a disclosing stain to interpret differences in biofilm formation. Biofilms were grown and assayed in triplicate or quadruplicate on three separate occasions (a total of *n* = 11 biofilms from three separate experiments). Data represented as mean ± SD. Statistical analysis performed using an unpaired Mann–Whitney t-test to compare differences in mean (*** *p* < 0.001).

**Figure 2 microorganisms-09-00059-f002:**
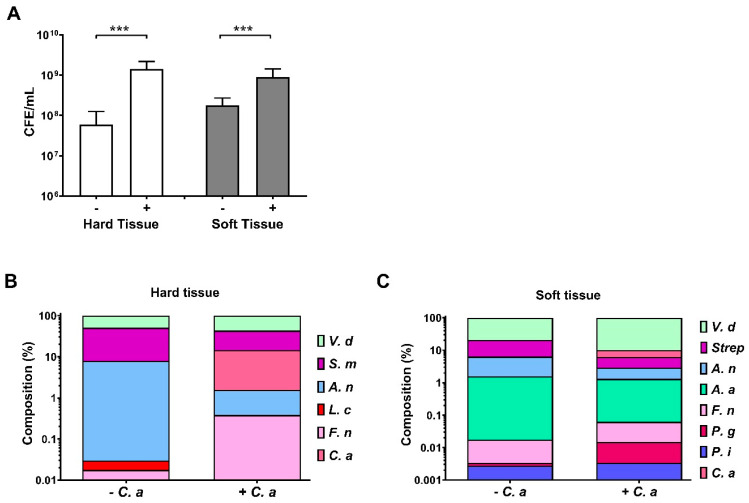
Bacterial load of hard tissue (HT) and soft tissue (ST) biofilms in the presence or absence *C. albicans*. Following growth and removal from the microtiter plate, DNA was extracted using the QIAamp DNA Mini Kit, and bacterial load assessed using 16s PCR. Fungal load was also determined using 18s PCR. The total bacterial load is shown in white bars for hard tissue and grey shaded bars for soft tissue (**A**). The 18s fungal counts were 4.4 × 10^7^ and 6.2 × 10^6^ CFE/mL for HT and ST biofilms, respectively. Further compositional analysis was carried out using species specific primers, as described in Table 1. HT biofilm composition is shown in (**B**) and ST biofilm composition represented in (**C**). Data represented as mean ± SD; n = 3 in three independent experiments. A Mann–Whitney t test was carried out to assess differences between means of bacterial load in biofilms containing *C. albicans* compared to without (*** *p* < 0.001).

**Figure 3 microorganisms-09-00059-f003:**
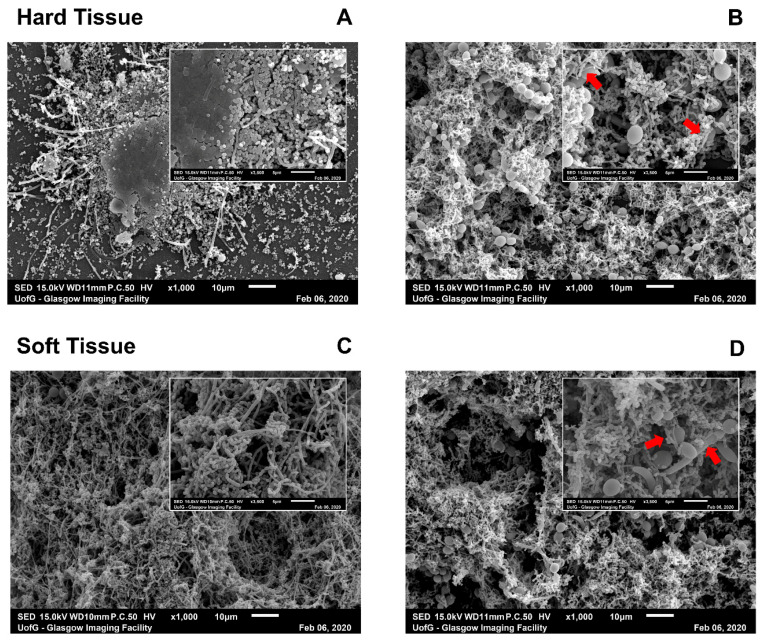
SEM images of hard tissue (HT) and soft tissue (ST) biofilms with or without *C. albicans*. Biofilms were processed after maturation and viewed on a JEOL JSM-6400 scanning electron microscope. Images were captured at ×1000 magnification (main image) and ×3500 magnification (inset). HT (**A**,**B**) and ST biofilms (**C**,**D**) were imaged without (**A**,**C**) and with *C. albicans* (**B**,**D**). Red arrows in insets of *Candida*-containing biofilms indicated visible attachment of bacterial cells with fungal hyphae or yeast cells. Images chosen were representative of duplicate samples.

**Figure 4 microorganisms-09-00059-f004:**
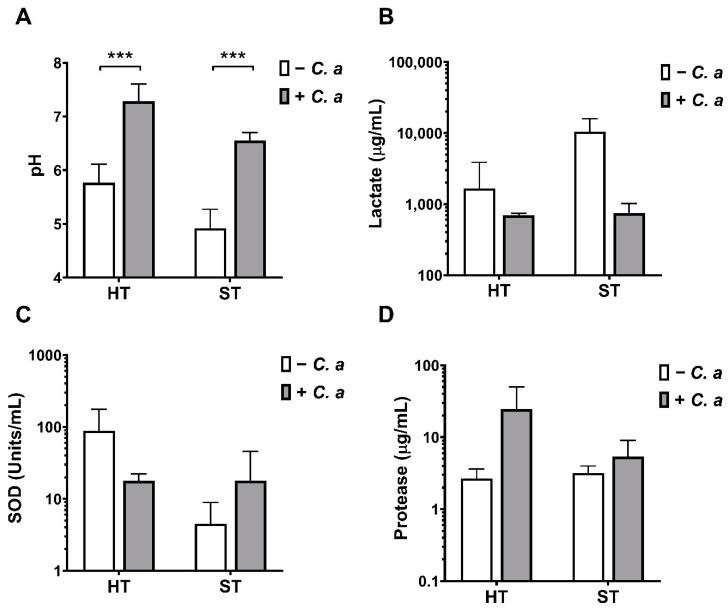
Assessment of functional biomarkers in hard tissue (HT) and soft tissue (ST) biofilms with or without *C. albicans*. Supernatants were collected on the final day of maturation and stored at −80° C until required. Prior to freezing, pH was tested using a microprobe (**A**). Lactate assay kit was used to determine lactate content (**B**). Superoxide dismutase assay was carried out using WST-1 tetrazolium salt and xanthine as per assay instructions with a 100× sample dilution applied (**C**). Protease activity was assessed using a fluorescent protease detection kit. Trypsin was used as a standard curve as per manufacturer’s instructions (**D**). Data represented as mean ± SD, sampled in triplicate from three independent experiments. A Mann–Whitney t-test was used throughout to assess significant differences between the means of models with or without *C. albicans* (*** *p* < 0.001).

**Figure 5 microorganisms-09-00059-f005:**
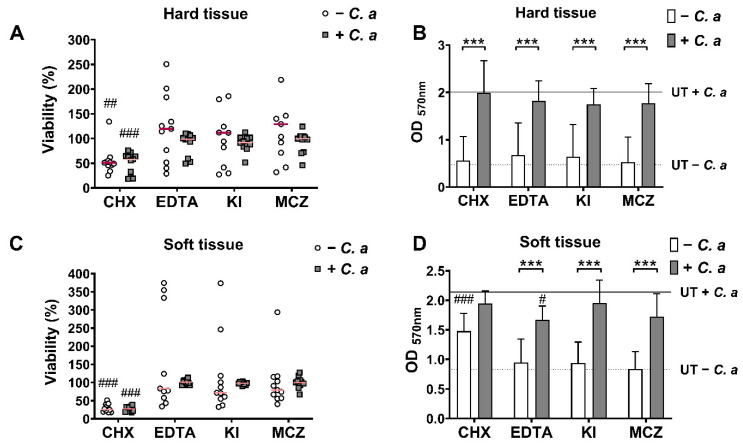
Treatment outcomes of hard and soft tissue biofilms with or without *C. albicans*. After maturation, biofilms were treated with 0.2% CHX, 4% EDTA, 1% KI or 40 µg/mL MCZ for 10 min at room temperature. Alamar blue™ dye was used to assess metabolic activity was calculated against an untreated control (HT biofilm (**A**) and ST biofilm (**C**)). Data represent mean ± SD, carried out in triplicate in three independent experiments. A Mann–Whitney t test was carried out on the raw metabolic activity data to assess significant differences between untreated (UT) and treated biofilms (## *p* < 0.01, ### *p* < 0.001). Following metabolic assessment, biofilms were dried overnight at room temperature and biomass was determined via crystal violet staining (HT biofilm (**B**) and ST biofilm (**D**)). A dotted black line has been drawn to illustrate the average results of the UT control—*C. albicans*, and a black solid line entered to show the UT control + *C. albicans*. Data represent mean ± SD, carried out in triplicate or quadruplicate on three independent experiments (a total of 11 replicates from 3 separate experiments). A Mann–Whitney t test was used to compare significant differences between UT biofilms and treated biofilms (# *p* < 0.05, ### *p* < 0.001, *** *p* < 0.001).

**Figure 6 microorganisms-09-00059-f006:**
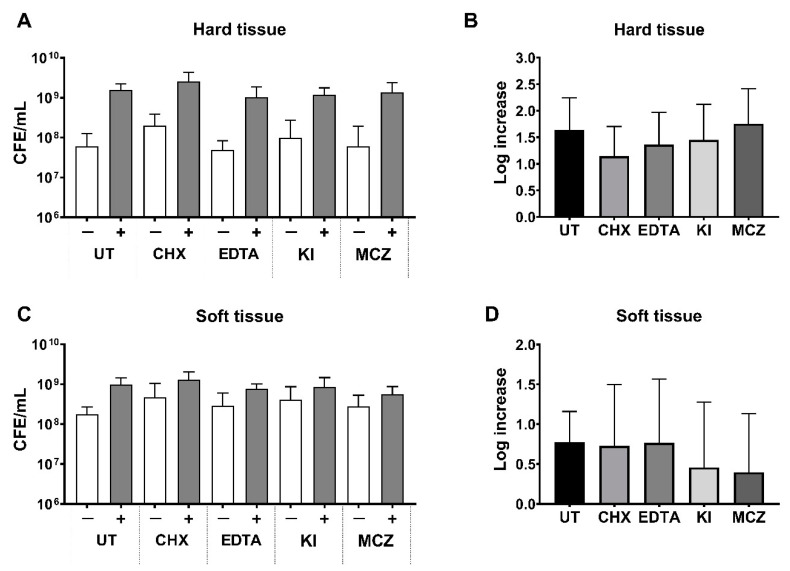
Bacterial load following 10 min treatment of hard and soft tissue biofilms plus or minus *C. albicans*. Following growth and removal of the biofilm from the microtiter plate, DNA was extracted using the QIAamp DNA Mini Kit, and bacterial load was assessed using 16S PCR. Data represent mean ± SD, experiment carried out in duplicate from three independent experiments. The total bacterial load (shown as colony forming equivalent (CFE) per mL) is shown for HT biofilms (**A**) and ST associated biofilms (**C**). The white bars are indicative of bacterial-only biofilms, whilst grey bars show the biofilms containing *C. albicans*. The fungal loads (quantified using 18s qPCR) of the biofilms were also assessed in the *C. albicans-*containing biofilms and total CFE/mL counts are documented in Table 3. Log increase, HT, caries associated biofilms (**B**) and ST associated biofilms (**D**) of biofilms with or without *C. albicans*. No significant differences were observed between untreated and treated samples using a Mann–Whitney t test.

**Table 1 microorganisms-09-00059-t001:** Primers used in the study.

Primer	Sequence (5′-3′)	References
*A. actinomycetemcomitans*	F-GAACCTTACCTACTCTTGACATCCGAA R-TGCAGCACCTGTCTCAAAGC	[44]
*A. naeslundii*	F-GGCTGCGATACCGTGAGG R-TCTGCGATTACTAGCGACTCC	[45]
*C. albicans*	F- CTCGTAGTTGAACCTTGGGC R- GGCCTGCTTTGAACACTCTA	[38]
*F. nucleatum* spp.	F-GGATTTATTGGGCGTAAAGC R-GGCATTCCTACAAATATCTACGAA	[46]
*P. intermedia*	F-CGGTCTGTTAAGCGTGTTGTG R-CACCATGAATTCCGCATACG	[44]
*P. gingivalis*	F-GGAAGAGAAGACCGTAGCACAAGGA R-GAGTAGGCGAAACGTCCATCAGGTC	[47]
*Streptococcus* spp.	F-GATACATAGCCGACCTGAG R-CCATTGCCGAAGATTCC	[46]
*V. dispar*	F-CCGTGATGGGATGGAAACTGC R-CCTTCGCCACTGGTGTTCTTC	[48]
*L. casei*	F-TGCACTGAGATTCGACTTAA R-CCCACTGCTGCCTCCCGTAGGAGT	[49]
16S	F-CGCTAGTAATCGTGGATCAGAATG R-TGTGACGGGCGGTGTGTA	[50]
18S	F-CTCGTAGTTGAACCTTGGGC R-GGCCTGCTTTGAACACTCTA	[51]

**Table 2 microorganisms-09-00059-t002:** Mean colony forming equivalent per mL and average percentage composition for each genus or species in the hard and soft tissue biofilms. CFE/mL and percentage composition shown for each genus or species as an average of 9 independent biofilms (n = 3 from 3 separate experiments). All numbers shown in the table have been rounded to the first significant decimal place, where possible.

Genus or Species	Mean CFE/mL and Percentage Composition (%)							
	Hard Tissue				Soft Tissue			
	− *C. albicans*		+ *C. albicans*		− *C. albicans*		+ *C. albicans*	
	CFE/mL	%	CFE/mL	%	CFE/mL	%	CFE/mL	%
***C. a***	n/a	n/a	4.60 × 10^7^	12.81	n/a	n/a	6.08 × 10^6^	4.00
***Strep***	7.12 × 10^6^	43.10	1.04 × 10^8^	29.00	2.87 × 10^6^	14.40	4.96 × 10^6^	3.26
***A. n***	1.29 × 10^6^	7.82	4.25 × 10^6^	1.18	9.50 × 10^5^	4.76	2.36 × 10^6^	1.55
***V. d***	8.10 × 10^6^	49.05	2.03 × 10^8^	56.63	1.58 × 10^7^	79.27	1.36 × 10^8^	89.87
***F. n***	2.91 × 10^3^	0.02	1.38 × 10^6^	0.38	2.80 × 10^3^	0.014	5.12 × 10^3^	0.05
***L. c***	1.99 × 10^3^	0.01	4.19 × 10^3^	0.001	n/a	n/a	n/a	n/a
***P. g***	n/a	n/a	n/a	n/a	1.23 × 10^2^	0.001	1.74 × 10^4^	0.011
***P. i***	n/a	n/a	n/a	n/a	5.49 × 10^2^	0.003	5.12 × 10^3^	0.003
***A. a***	n/a	n/a	n/a	n/a	3.11 × 10^5^	1.56	1.90 × 10^6^	1.25

**Table 3 microorganisms-09-00059-t003:** Average colony forming equivalent counts for total bacteria and fungi in the two biofilm models following treatment. The mean CFE/mL for each biofilm was calculated from an average of 9 independent biofilms (n = 3 from 3 separate experiments) following treatment with CHX, EDTA, KI and MCZ compared to untreated controls.

Biofilm	Treatment	Average CFE/mL			
		16S		18S	
		− *C. a*	+ *C. a*	− *C. a*	+ *C. a*
ST	UT	1.77 × 10^8^	9.96 × 10^8^	n/a	6.08 × 10^6^
	CHX	4.74 × 10^8^	1.30 × 10^9^	n/a	4.31 × 10^6^
	EDTA	2.91 × 10^8^	7.66 × 10^8^	n/a	3.91 × 10^6^
	KI	4.13 × 10^8^	8.46 × 10^8^	n/a	3.32 × 10^6^
	MCZ	2.79 × 10^8^	5.60 × 10^8^	n/a	3.21 × 10^6^
HT	UT	6.00 × 10^7^	1.59 × 10^9^	n/a	4.60 × 10^7^
	CHX	2.02 × 10^8^	2.59 × 10^9^	n/a	4.21 × 10^7^
	EDTA	4.91 × 10^7^	1.04 × 10^9^	n/a	3.34 × 10^7^
	KI	9.84 × 10^7^	1.18 × 10^9^	n/a	3.11 × 10^7^
	MCZ	6.13 × 10^7^	1.36× 10^9^	n/a	3.52 × 10^7^

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
