# Peer review of "Candida albicans as an Essential “Keystone” Component within Polymicrobial Oral Biofilm Models?"

_microorganisms, 2020, doi:10.3390/microorganisms9010059_

Round 1
Reviewer 1 Report
Candida albicans as an essential ”keystone” component within polymicrobial oral biofilm models?
Tracy Young et al.,
This study describes a multispecies biofilm models including several known oral bacteria and the effects of Candida albicans on those biofilms. Authors have created two different types of polymicrobial biofilms, one representing biofilms on hard tissues in the oral cavity and the other containing species representing bacteria found on oral soft tissues. Biofilms are formed with and without Candida albicans and the effect of C. albicans inclusion on microbial composition, metabolic activity and formed biofilm mass are evaluated. Acid production as well as production of reactive oxygen species, ROS, are also measured. Finally, biofilms are visualized by scanning electron microscopy. To test the susceptibility of different biofilms on different antimicrobial treatments, the biofilms were exposed to chlorhexidine gluconate, EDTA, KI or MCZ.
The manuscript is well written and easy to follow. The study models are elegant and contain relevant microbial components to represent different types of oral biofilms. The methods used to evaluate different aspects of the biofilms are established and widely used methods in biofilm studies and thus well suited in the analysis. The results describing the effects of C. albicans on the biofilms are presented in adequate figures 1-4 and tables 1-2. Yet, the conclusion made in the end of abstract and discussion, that “…data support the conclusion that yeasts are potential ‘keystone’ components of oral biofilms” or “… C. albicans may serve as a keystone commensal” can be considered as a slight over estimation of the data. The data show that C. albicans modifies the biofilm composition, but does it make it ‘a potential keystone component’? Wouldn’t it be likely that any addition of a new component to the used model would have effects on measured parameters?
The biggest problem of the manuscript is, however, the part of biofilm treatment with the antimicrobials. In the Materials&Methods section there is no description of how the treatments were made. How long the biofilms were exposed to the substances? How were the substances added on the biofilm? Were they added in the medium or in buffer? Etc. This should be clearly stated in M&M section, to make it possible to evaluate the obtained results.
In the figure 5 legend it is said that “… biofilms were treated …. for 5 mins.” If that is how the treatments were done, then one can ask what is the relevance of measuring biofilm mass or microbial composition with qPCR after such a short exposure? With qPCR one measures both live and dead bacteria, so can one really expect to see a difference in these results after 5 min exposure? Also, the biofilm mass, as measured by crystal violet, is rather crude parameter, and unless it is expected that some of the antimicrobial components would dissolve the biofilm matrix, it is difficult to see the sense behind measuring it.
Reviewer would like to believe that the treatments were done in a different way, and describing it in the M&M section would clarify the issue. If so, there is still a problem in the comparisons made in figures 5&6. The authors have calculated, for example, differences between biofilm mass of HT biofilm with and without candida, and found a significant difference. The biofilm mass of biofilms with and without C. albicans is shown to be different already in the Fig 1B. In the figures 5&6 the authors are presenting results of the ANTIMICROBIAL TREATMENTS of the biofilms. There the only relevant comparison is between treated and untreated biofilms. This is correctly done in Figure 5 A and C. Please, correct that to the other parts as well. And modify results and discussion sections accordingly. The effect of candida can, of course, be discussed, but at this point in the treatment susceptibility point of view.
Additional small points:
p.4 line 159 it is written that bacteria were added in 96 well flat bottom plates (ThermoFisher, UK), but earlier on the same p. line 152 it was stated that 96 well plates were from Corning,UK. Please, correct.
p.4. line 161. Define AS
p.4. line 192 it is written that …colour changes were measured when growth control reached a sufficient pink…, please define “sufficient”
p.5. line 202 define CV when it is first time used
p.5. 215 . It is written …bacterial biofilms were removed by using pipette tip… How was it confirmed that all bacteria in biofilm was included in the analysis and no significant amount of bacteria remained attached on the plate surface?
p.5. line 221 reference source missing, please correct
p.6. 234 it is said that biofilms grown on 13mm2 coverslips were used. It is not described how were these biofilms made. Please, add. Line 235 it is stated that “biofilms were carefully washed”. According to “minimum information guidelines specified for biofilm formation” that the authors say they are following such expressions should not be used. Please, specify washing conditions.
- 6 line 254 it is expressed that tubes were centrifuged 10,000 rpm, please use g-values. Please, check the whole section and correct accordingly.
- 7 lines 278 – 286, please specify the treatment protocol
Results section general: when significances are expressed in the text, the p-value is enough. Please, remove the stars. E.g. p. 7 line 299 ***p<0.001 should be p<0.001
- 12 line 393 it is stated “pH of both biofilms…” The pH of the medium was measured, not that of biofilms. The pH of a biofilm can be very different from that of the medium. Please correct accordingly to the whole paragraph.
- 13 line 422 starts “Table 5 shows…” There is no table 5 in the manuscript. Please, correct.
- 13 lines 422-442, please clarify the results in line with comments above.
Figure 1 In figure legend it is said that experiments were done in quadruplicates and repeated three times, still only 11 dots are visible in all groups. Why is that?
Figure 2 Use letters A, B, C instead of A, Bi, Bii
Figure 3 Use letters A, B, C, D instead of Ai, Aii, Bi, Bii
Figure 4 Is that correct, that the only significant difference was wound in pHs of growth medium? In the figure legend it is indicated **p<0.01, but in the figure there is only *** marked. Please, use uniform marks in the legend and figure.
Author Response
Response to Reviewer 1 – Candida albicans as an essential ‘keystone’ component within polymicrobial oral biofilm models?
Firstly, the authors would like to thank reviewer 1 for their insightful and supportive comments. We have considered these and where appropriate made changes to the manuscript (tracked changes), or provided a detailed response point-by-point to the reviewer(s) comments below in red. We believe that by addressing these comments we have been able to significantly improve our manuscript, and hope the revised article is now suitable for acceptance and publication in Microorganisms journal.
This study describes a multispecies biofilm models including several known oral bacteria and the effects of Candida albicans on those biofilms. Authors have created two different types of polymicrobial biofilms, one representing biofilms on hard tissues in the oral cavity and the other containing species representing bacteria found on oral soft tissues. Biofilms are formed with and without Candida albicans and the effect of C. albicans inclusion on microbial composition, metabolic activity and formed biofilm mass are evaluated. Acid production as well as production of reactive oxygen species, ROS, are also measured. Finally, biofilms are visualized by scanning electron microscopy. To test the susceptibility of different biofilms on different antimicrobial treatments, the biofilms were exposed to chlorhexidine gluconate, EDTA, KI or MCZ.
The manuscript is well written and easy to follow. The study models are elegant and contain relevant microbial components to represent different types of oral biofilms. The methods used to evaluate different aspects of the biofilms are established and widely used methods in biofilm studies and thus well suited in the analysis. The results describing the effects of C. albicans on the biofilms are presented in adequate figures 1-4 and tables 1-2. Yet, the conclusion made in the end of abstract and discussion, that “…data support the conclusion that yeasts are potential ‘keystone’ components of oral biofilms” or “… C. albicans may serve as a keystone commensal” can be considered as a slight over estimation of the data. The data show that C. albicans modifies the biofilm composition, but does it make it ‘a potential keystone component’? Wouldn’t it be likely that any addition of a new component to the used model would have effects on measured parameters?
We are grateful that the reviewer believes that “the manuscript is well written, easy to follow and the studied models are elegant and contain relevant microbial components to represent different types of oral biofilms …”. In hindsight, we agree that out conclusions regarding C. albicans being a keystone commensal are slightly over-estimated in regard to the current data. Where appropriate (in particular the discussion section), we have attempted to weaken the conclusions around the idea that this fungal species may serve as a keystone commensal, at least in reference to the data presented in this paper. At this juncture, we would like to direct the reviewer to a paper by Janus et al in 2016 (Adv Exp Med Biol. 2016;931:13-20) who were the first to propose this concept. We merely wanted to test the hypothesis that C. albicans may serve as keystone commensal within our model systems and present the results accordingly. We have attempted to clarify this were possible and state that is not our hypothesis per se, and that we were just testing the hypothesis.
Regarding the reviewers’ second statement, we partially agree in that inclusion of any fungal element (albeit strain-dependent), and possibly some bacterial species, would affect the measured parameters. Again, we have emphasized this in the passage, speculating that other fungal species may serve as keystone commensals simply due to their ability to adjust environmental conditions and by providing a scaffold for bacterial adhesion.
The biggest problem of the manuscript is, however, the part of biofilm treatment with the antimicrobials. In the Materials&Methods section there is no description of how the treatments were made. How long the biofilms were exposed to the substances? How were the substances added on the biofilm? Were they added in the medium or in buffer? Etc. This should be clearly stated in M&M section, to make it possible to evaluate the obtained results.
We have amended the material and methods section accordingly and have covered all the points raised by the reviewer. This section starts at line 307, page 7 in the article with tracked changes.
In the figure 5 legend it is said that “… biofilms were treated …. for 5 mins.” If that is how the treatments were done, then one can ask what is the relevance of measuring biofilm mass or microbial composition with qPCR after such a short exposure? With qPCR one measures both live and dead bacteria, so can one really expect to see a difference in these results after 5 min exposure? Also, the biofilm mass, as measured by crystal violet, is rather crude parameter, and unless it is expected that some of the antimicrobial components would dissolve the biofilm matrix, it is difficult to see the sense behind measuring it.
We appreciate the reviewers concerns regarding the methods used for such short-term incubations. We have amended the methods section and included details of the treatment process used (line 307, page 7).
Regarding the qPCR methodology, we have included a statement in the discussion that references previous publications that show short term incubations of the biofilms with therapeutics (e.g., 5-10 mins) such as CHX can reduce the viability, which hypothetically could lead to detachment (Zhou et al., Biomed Res Int. 2019 Jan 21;2019:6393470; Shen et al., Sci Rep. 2016; 6: 27537. and Hope & Wilson., Antimicrob Agents Chemother. 2004 May;48(5):1461-8.).
We agree that the CV measurement is a very crude and basic measurement for such studies and highlighted such flaws in this methodology in the original manuscript discussion. However, there is some evidence in the literature that suggests such treatments (e.g., EDTA) can dissolve the matrix, thereby impacting the biomass readings (Cavaliere et al., Microbiology open. 2014 Aug; 3(4): 557–567). Therefore, we believed it was necessary to assess the biomass within the context of the treatment study. Again, we have stated this in the revised discussion.
Reviewer would like to believe that the treatments were done in a different way, and describing it in the M&M section would clarify the issue. If so, there is still a problem in the comparisons made in figures 5&6. The authors have calculated, for example, differences between biofilm mass of HT biofilm with and without candida, and found a significant difference. The biofilm mass of biofilms with and without C. albicans is shown to be different already in the Fig 1B. In the figures 5&6 the authors are presenting results of the ANTIMICROBIAL TREATMENTS of the biofilms. There the only relevant comparison is between treated and untreated biofilms. This is correctly done in Figure 5 A and C. Please, correct that to the other parts as well. And modify results and discussion sections accordingly. The effect of candida can, of course, be discussed, but at this point in the treatment susceptibility point of view.
We agree with the reviewer that the only relevant analyses for these figures should be +/- antimicrobial treatment, and not for drawing comparisons between treated biofilms +/- candida. As such, we have removed any discussions relating to statistical analyses on the figures to treatments +/- candida and focused our description of results to treatments alone.
Additional small points:
p.4 line 159 it is written that bacteria were added in 96 well flat bottom plates (ThermoFisher, UK), but earlier on the same p. line 152 it was stated that 96 well plates were from Corning,UK. Please, correct.
This has been amended.
p.4. line 161. Define AS
This was a typo and should read as “RPMI/THB media”. This has been corrected.
p.4. line 192 it is written that …colour changes were measured when growth control reached a sufficient pink…, please define “sufficient”
We apologize for this ambiguity. We have changed this to “bright pink colour”.
p.5. line 202 define CV when it is first time used
A definition for CV has been included here.
p.5. 215 . It is written …bacterial biofilms were removed by using pipette tip… How was it confirmed that all bacteria in biofilm was included in the analysis and no significant amount of bacteria remained attached on the plate surface?
We thank the reviewers for indicating this oversight in regard to our methodology. We have included in the revision how we assessed that no biomass remained in the plates.
p.5. line 221 reference source missing, please correct
The correct references have been included.
p.6. 234 it is said that biofilms grown on 13mm2 coverslips were used. It is not described how were these biofilms made. Please, add. Line 235 it is stated that “biofilms were carefully washed”. According to “minimum information guidelines specified for biofilm formation” that the authors say they are following such expressions should not be used. Please, specify washing conditions.
The authors would like to apologize for this ambiguity for the methodology and agree that such terminology should be avoided if correctly adhering to the minimum guidelines for biofilm formation. We have amended this sentence and proofed the rest of the manuscript to ensure all other steps have been correctly described in detail, in accordance to the above guidelines.
Page 6 line 254 it is expressed that tubes were centrifuged 10,000 rpm, please use g-values. Please, check the whole section and correct accordingly.
All references to rpm values have been corrected to g-values.
Page 7 lines 278 – 286, please specify the treatment protocol
A detailed protocol for the treatment element of the study has been included here on page 7.
Results section general: when significances are expressed in the text, the p-value is enough. Please, remove the stars. E.g. p. 7 line 299 ***p<0.001 should be p<0.001
Page 12 line 393 it is stated “pH of both biofilms…” The pH of the medium was measured, not that of biofilms. The pH of a biofilm can be very different from that of the medium. Please correct accordingly to the whole paragraph.
A very good point made re. the pH of the biofilm being different to the pH of the spent media. As such we have corrected this throughout the paragraph
Page 13 line 422 starts “Table 5 shows…” There is no table 5 in the manuscript. Please, correct.
We apologize for this oversight, this should be in reference to Figure 5 and table 3.
Page 13 lines 422-442, please clarify the results in line with comments above.
The results section for Figure 5 and 6 have been amended to ensure results are discussed in relationship to treatment only within their respective +/- candida groups.
Figure 1 In figure legend it is said that experiments were done in quadruplicates and repeated three times, still only 11 dots are visible in all groups. Why is that?
We thank the reviewer for their attention to detail. We have amended the methods as they have should originally stated that the experiments were done in either triplicate or quadruplicate. We have also stated, where possible, the total number of replicates per study.
Figure 2 Use letters A, B, C instead of A, Bi, Bii
This has been amended in the revised figures.
Figure 3 Use letters A, B, C, D instead of Ai, Aii, Bi, Bii
This has been amended in the revised figures.
Figure 4 Is that correct, that the only significant difference was wound in pHs of growth medium? In the figure legend it is indicated **p<0.01, but in the figure there is only *** marked. Please, use uniform marks in the legend and figure.
Yes, the only significance difference was observed between the pH values. We have changed the legend so that the asterisks in panel A match with the legend.
Reviewer 2 Report
The manuscript by Young and co-authors describes experiments with polymicrobial biofilms and the effect of including Candida albicans in such consortia. Overall, the study is well-conducted and –described. The study highlights the importance of studying polymicrobial communities for making relevant conclusions and defines a reproducible model for oral biofilms.
Suggestions for improvement (numbers refer to line numbers):
The title could be improved. Why is it formulated as a question? Why not “... is an essential …"?
19: “.... assays focus on …"
21: relevant instead of “reproducible complex”?
25: No hyphens
27: estimated instead of “established”?
29: Here and on many occasions in the remainder of the manuscript: pH and lactate are not bioassays! It should only be stated that pH and lactate were measured as biomarkers for virulence.
31: by more than a factor of 10 in the presence of …
32: Here and throughout the results and discussion sections: Own results must be reported in the past tense! …. I would suggest sho that inclusion of …. impacted ….
45: and instead of with and defines instead of defining
50: Sentence starting with “Specificity …" is confusing and not clear at all.
69: hyphae
72: 2 spaces before “F.” … delete “of”, hyphae, comma after “albicans”
76-77: A “keystone commensal” seems a contradiction in itself and should be avoided. A commensal is defined as an organism not affecting its “host” …. how could it be a keystone species then?
95: Here and elsewhere: The symbol ± should be avoided in text section – please write with words (with and without, in the presence and absence of …). Suggestion for simplification: “... of the biofilms and biofilm response to conventional therapeutics will be assessed in the presence and absence of C. albicans.”
97: fungus
102: were instead of was
122: chambers were
130: Should mM mean millimeter? This is confusing and certainly not the standard abbreviation; please use mm.
139: comma after previously
140: the before Miles
148: “similar manner as previously …"
150: was instead of is
153: were instead of was
154: delete “removing and replacing media”
160: was instead of were
170:was instead of were
173: “medium was replaced …"
174: delete “then”
178: Not clear; what profile?
181: Suggestion: “Changes in pathogenesis were also estimated by measuring fluctuations in ….”
190: replace “by via” with involved in
192: sufficiently pink color … or the latest after 3 h, ….
194: Here and elsewhere: always put a space between number and unit.
221: Missing reference
222: Missing p
231: make the table wider, the species name should not be split on three lines.
252: Not bioassays!
264: New sentence after solution
266: according to the instead of upon
281-282: Suggestion: Chlorhexidine … is a gold standard used within the dentistry industry and was used to mimic standard daily mouth washing.
283: Maybe irritant instead of irrigant?
284: served after 1 %
294ff: Throughout the results: Pay close attention to past/present tense – own results should be reported in the past tense!
295ff: The first section of the results, presented in Figure 1, seems to miss a control. As discussed, it seems obvious that the biomass increases if an additional species is present. The control with C. albicans alone is missing. It would also be helpful to show a time-course so that it is possible to appreciate the growth of the biofilm over time. As including such controls may be beyond the scope here, the first section should probably be combined with the second one, as things become clearer here.
310: C. albicans in italics
328: Figure 2A: It is not clear why the white and shaded bars are ontop of each other. Does this mean that the incease is only due to Candida (I.e., the shaded bar)? This must be explained and clarified. Since the two bars show two different things (16S and 18S data), they should not be ontop of each other.
365: Table 2: The numbers should be rounded to the first significant decimal.
369ff: Past tense!
389: Not bioassays!
390: have instead of has; established instead of eluded?; delete to
398ff: Suggestion: The modulation of lactate production by microbial consortia is a common mechanism ….
422: Table 3 (?) shows …
585: delete “able to” and maybe an alternative for produce? Cause?
587: It should be explained what is meant by morphogenesis.
Author Response
Response to Reviewer 1 – Candida albicans as an essential ‘keystone’ component within polymicrobial oral biofilm models?
We are grateful for comments provided by reviewer 2. We have considered these and where appropriate made changes to the manuscript (tracked changes), or provided a detailed response point-by-point to the reviewer(s) comments below in red. We hope the reviewer will approve of our changes and accept that the manuscript is suitable for publication in Microorganisms journal.
The manuscript by Young and co-authors describes experiments with polymicrobial biofilms and the effect of including Candida albicans in such consortia. Overall, the study is well-conducted and –described. The study highlights the importance of studying polymicrobial communities for making relevant conclusions and defines a reproducible model for oral biofilms.
We are extremely appreciative of the reviewers’ kind words re. our manuscript and delighted that they found the “study was well-conducted and described”. The authors are hopeful future model systems consider the incorporation of a fungal element to truly mimic the oral polymicrobial communities.
Suggestions for improvement (numbers refer to line numbers):
The title could be improved. Why is it formulated as a question? Why not “... is an essential …"?
We are happy to change title at the reviewers and/or editors discretion but feel such a change would potentially over-estimate the results from the study. Unlike the “keystone pathogen hypothesis” publication (for example) which unequivocally shows that the P. gingivalis acts as a keystone pathogen, the data shown here can be interpreted in different ways. For instance, there were no changes to the biofilm tolerance to treatment following inclusion of C. albicans, which to some of the readership, would be the most important “measure” of the organism being a keystone. Alternatively, to others, a thicker biofilm with increases in bacterial bioburden and/or changes in virulence markers would in fact show the inclusion of a fungal element warrants it as consideration as a keystone component.
19: “.... assays focus on …"
This has been amended.
21: relevant instead of “reproducible complex”?
This has been altered accordingly.
25: No hyphens
The hyphens have been removed.
27: estimated instead of “established”?
The word established has been changed to estimated.
29: Here and on many occasions in the remainder of the manuscript: pH and lactate are not bioassays! It should only be stated that pH and lactate were measured as biomarkers for virulence.
We apologize for the incorrect nomenclature used throughout the paper. The whole paper has been proofed and all these errors have been corrected.
31: by more than a factor of 10 in the presence of …
This has been amended.
32: Here and throughout the results and discussion sections: Own results must be reported in the past tense! …. I would suggest sho that inclusion of …. impacted ….
The whole passage had been proofread for such errors in tense.
45: and instead of with and defines instead of defining
50: Sentence starting with “Specificity …" is confusing and not clear at all.
69: hyphae
72: 2 spaces before “F.” … delete “of”, hyphae, comma after “albicans”
The above four alterations have been made as suggested by the reviewer.
76-77: A “keystone commensal” seems a contradiction in itself and should be avoided. A commensal is defined as an organism not affecting its “host” …. how could it be a keystone species then?
We want to emphasize that this terminology was postulated elsewhere (Janus et al, Adv Exp Med Biol. 2016;931:13-20) and we are merely testing the theory, which provides the premise for the paper (as shown by the question in the title). As theorized in the above paper, one could argue that keystone species are those that are present in low numbers although have a considerable effect on the ecology of the microenvironment (eg., in the oral cavity where Candida is present in low cell number). Even as a commensal we feel the importance of Candida is evident due to its interactions with many different bacteria. Therefore, we feel the term “keystone” can be considered valid. As such, with all due respect, we feel the term should still be included where appropriate.
95: Here and elsewhere: The symbol ± should be avoided in text section – please write with words (with and without, in the presence and absence of …). Suggestion for simplification: “... of the biofilms and biofilm response to conventional therapeutics will be assessed in the presence and absence of C. albicans.”
Where appropriate, the symbol “±” has been removed throughout the passage.
97: fungus
102: were instead of was
122: chambers were
130: Should mM mean millimeter? This is confusing and certainly not the standard abbreviation; please use mm.
139: comma after previously
140: the before Miles
148: “similar manner as previously …"
150: was instead of is
153: were instead of was
154: delete “removing and replacing media”
160: was instead of were
170:was instead of were
173: “medium was replaced …"
174: delete “then”
All the above changes (from line 97 to line 174) have been amended in the revised article.
178: Not clear; what profile?
We apologize for the confusion regarding this sentence. The new sentence reads “to establish the importance of incorporating C. albicans into the biofilms…”
181: Suggestion: “Changes in pathogenesis were also estimated by measuring fluctuations in ….”
This has been changed.
190: replace “by via” with involved in
“by via” has been replaced.
192: sufficiently pink color … or the latest after 3 h, ….
This has been amended.
194: Here and elsewhere: always put a space between number and unit.
The whole manuscript has been proofed for such errors.
221: Missing reference
The correct references have been included here.
222: Missing p
This has been corrected.
231: make the table wider, the species name should not be split on three lines.
The table has been made wider.
252: Not bioassays!
The use of the word bioassay has been changed throughout the passage.
264: New sentence after solution
266: according to the instead of upon
281-282: Suggestion: Chlorhexidine … is a gold standard used within the dentistry industry and was used to mimic standard daily mouth washing.
The above three corrections have been made.
283: Maybe irritant instead of irrigant?
The use of the word irrigant is correct here. EDTA is often referred to a root canal “irrigating solution” in endodontic treatment.
284: served after 1 %
This has been changed.
294ff: Throughout the results: Pay close attention to past/present tense – own results should be reported in the past tense!
The whole manuscript has been proofread for such errors. We thank the reviewer for indicating this.
295ff: The first section of the results, presented in Figure 1, seems to miss a control. As discussed, it seems obvious that the biomass increases if an additional species is present. The control with C. albicans alone is missing. It would also be helpful to show a time-course so that it is possible to appreciate the growth of the biofilm over time. As including such controls may be beyond the scope here, the first section should probably be combined with the second one, as things become clearer here.
Inclusion of a C. albicans only control would not be useful for these comparative studies, given that C. albicans mono-species biofilms are usually only grown for 24-48 hours max. Therefore, it would likely not be applicable given that the mixed species models are grown for up to 8 days.
Regarding the time-course experiment, we agree this would provide growth kinetics for the mixed species biofilms. The models used here have been published elsewhere (Sherry et al. Front Microbiol, 2016. 7: p. 912; Brown et al., Scientific Reports, 2019. 9(1): p. 1577 and Zhou, Y., et al., Front Microbiol, 2018. 9: p. 1498), and therefore were selected as “established” biofilm systems. Due to the costs attached to repeating such experiments with time-course intervals for all experimental outputs, it is unlikely this is feasible, and we don’t believe this would improve our understanding of importance of Candida in the such models.
As per the reviewers’ request, we have combined section 1 and 2 so that the metabolic activity, CV data and compositional analyses are all in the same section.
310: C. albicans in italics
This has been corrected.
328: Figure 2A: It is not clear why the white and shaded bars are ontop of each other. Does this mean that the incease is only due to Candida (I.e., the shaded bar)? This must be explained and clarified. Since the two bars show two different things (16S and 18S data), they should not be ontop of each other.
We have clarified in the legends of Figure 2 and Figure 6 the use of the white and grey bars in these figures. In the interest of space (especially in figure 6) the 16s and 18s data has been incorporated into one bar. We feel this also nicely shows the proportion of candida (18S) within the mixed species biofilm (e.g., it is present but in much lower proportions than the total microbial load).
Regarding the question “is the increase only due to the candida?” – no, Candida is there in much lower proportions in the mixed species biofilm therefore only takes up part of the bar. For example, in HT figure 2A, the values for the CFE/mL loads are as follows; bacterial load of 6×107 CFE/mL minus Candida, which increased to 2×109 CFE/mL when C. albicans was included (statistically significant as indicated by the ***). The CFE/mL counts for Candida were 5×107 CFE/mL. Given that Candida is only present in one of the biofilms for HT and ST, having this on a separate graph would be redundant as we are not comparing 18S counts between HT and ST biofilms.
365: Table 2: The numbers should be rounded to the first significant decimal.
369ff: Past tense!
389: Not bioassays!
390: have instead of has; established instead of eluded?; delete to
398ff: Suggestion: The modulation of lactate production by microbial consortia is a common mechanism ….
422: Table 3 (?) shows …
585: delete “able to” and maybe an alternative for produce? Cause?
The above suggestions (line 365 – line 585) have all been amended in the revised passage.
587: It should be explained what is meant by morphogenesis.
A short definition has been included for morphogenesis.
Round 2
Reviewer 1 Report
Tha manuscript has significantly improved by the modification made by the authors. There are, however, still few points that needs attention.
Firstly, in results section p. 14-15 and in Figure 5 as well as in figure legend 5 it is described that "viability" of bacteria was measured, but the authors measured metabolic activity. Please, correct that. Correct that also to discussion, e.g. line 652.
The second point relates to qPCR method used to evaluate the effect of the short-term antimicrobial treatments. As pointed out already in previous comments qPCR does not differentiate between live and dead bacteria. The authors argue that dead bacteria would detach from biofilm, but from mature biofilm that is not likely to happen in significant amounts in such a short time. Authors provide some reference relating to the matter (Zhou et al., Biomed Res Int. 2019 Jan 21;2019:6393470; Shen et al., Sci Rep. 2016; 6: 27537. and Hope & Wilson., Antimicrob Agents Chemother. 2004 May;48(5):1461-8.) and at least two of those references that were available in pub med, also clearly show that dead bacteria remain in the biofilm. This is evident also from many other studies. Thus, the qPCR results should be clearly discussed in that aspect, or they should be left out from the manuscript.
Author Response
Tha manuscript has significantly improved by the modification made by the authors. There are, however, still few points that needs attention.
We are grateful the reviewer believes the manuscript has significantly improved following the first revision. We have amended the manuscript again in line with reviewers comments, and have highlighted all changes in the resubmitted version.
Firstly, in results section p. 14-15 and in Figure 5 as well as in figure legend 5 it is described that "viability" of bacteria was measured, but the authors measured metabolic activity. Please, correct that. Correct that also to discussion, e.g. line 652.
Any reference to "viability" of the biofilm has been changed to "metabolic activity", where appropriate.
The second point relates to qPCR method used to evaluate the effect of the short-term antimicrobial treatments. As pointed out already in previous comments qPCR does not differentiate between live and dead bacteria. The authors argue that dead bacteria would detach from biofilm, but from mature biofilm that is not likely to happen in significant amounts in such a short time. Authors provide some reference relating to the matter (Zhou et al., Biomed Res Int. 2019 Jan 21;2019:6393470; Shen et al., Sci Rep. 2016; 6: 27537. and Hope & Wilson., Antimicrob Agents Chemother. 2004 May;48(5):1461-8.) and at least two of those references that were available in pub med, also clearly show that dead bacteria remain in the biofilm. This is evident also from many other studies. Thus, the qPCR results should be clearly discussed in that aspect, or they should be left out from the manuscript.
We appreciate the reviewers concern with regard to the qPCR methodology, and the inability of the technique to differentiate between live and dead organisms. The authors apologize for this oversight. We have added further discussion in the final section of the paper (page 20, line 639 onwards) to fully clarify that the methodology used in this study has its limitations, and indeed, as astutely stated by the reviewer, that dead organisms may persist in the biofilm following treatment.
Reviewer 2 Report
This reviewer appreciates the changes and amendments that were made by Young and colleagues. For the final and to be published version of the manuscript it might nevertheless be nice if another solution was found for the data in 2A and 6A.
The problem is the following: Since the two columns for the “+” treatment are on top of each other, it is not clear if they add up or are actually two separate columns that are just on top – as it is the case. This is why people may think that the increase is only due to the presence of Candida, as the white part of the combined column is about as high as it is in the “-“ treatment. In particular in Figure 2 this impression may be obtained, because the columns in the panels B and C are composites, where each color only corresponds to a fraction of the entire bar. At the very least, this should be explained in the Figure legend; maybe by including the explanation the authors provided in the responses (indicating the exact values for the bacterial and Candida cell densities in the different treatments).
Author Response
This reviewer appreciates the changes and amendments that were made by Young and colleagues. For the final and to be published version of the manuscript it might nevertheless be nice if another solution was found for the data in 2A and 6A.
The problem is the following: Since the two columns for the “+” treatment are on top of each other, it is not clear if they add up or are actually two separate columns that are just on top – as it is the case. This is why people may think that the increase is only due to the presence of Candida, as the white part of the combined column is about as high as it is in the “-“ treatment. In particular in Figure 2 this impression may be obtained, because the columns in the panels B and C are composites, where each color only corresponds to a fraction of the entire bar. At the very least, this should be explained in the Figure legend; maybe by including the explanation the authors provided in the responses (indicating the exact values for the bacterial and Candida cell densities in the different treatments).
We thank the reviewer for their follow up comments and have amended the manuscript accordingly. Specifically, Figure 2A and Figure 6A/C have been changed in that the fungal 18S counts have been removed to avoid confusion. Reference to the 18S counts have been made in the figure legends. The authors hope that such amendments have improved the clarity of the data, clearly showing that bacterial counts increase following inclusion of Candida spp.